

# Persistent microbial dysbiosis in preterm premature rupture of membranes from onset until delivery

Elizabeth A. Baldwin[1,*], Marina Walther-Antonio[2,*], Allison M. MacLean[3], Daryl M. Gohl[3], Kenneth B. Beckman[3], Jun Chen[4], Bryan White[5,6], Douglas J. Creedon[7,*] and Nicholas Chia[2,8,*]

[1] Department of Maternal Fetal Medicine, Mayo Clinic, Rochester, MN, USA
[2] Department of Surgical Research, Mayo Clinic, Rochester, MN, USA
[3] Genomics Center, University of Minnesota, Minneapolis, MN, USA
[4] Biomedical Statistics and Informatics, Mayo Clinic, Rochester, MN, USA
[5] Carl R. Woese Institute for Genomic Biology, University of Illinois at Urbana-Champaign, Urbana, IL, United States
[6] Animal Sciences, University of Illinois at Urbana-Champaign, Urbana, IL, United States
[7] Obstetrics and Gynecology, Mayo Clinic, Rochester, MN, USA
[8] Department of Physiology and Biomedical Engineering, Mayo Clinic, Rochester, MN, USA
* These authors contributed equally to this work.

Corresponding author
Marina Walther-Antonio,
waltherantonio.marina@mayo.edu

## ABSTRACT

**Background.** Preterm Premature Rupture of Membranes (PPROM) is a major leading cause of preterm births. While the cause for PPROM remains unidentified, it is anticipated to be due to subclinical infection, since a large proportion of PPROM patients display signs of chorioamnionitis. Since subclinical infections can be facilitated by dysbiosis, our goal was to characterize the vaginal microbiome and amniotic fluid discharge upon PPROM, through latency antibiotic treatment, and until delivery, to detect the presence of pathogens, microbiota alteration, and microbial response to treatment.

**Methods.** Enrolled subjects (15) underwent routine institutional antenatal care for PPROM, including the administration of latency antibiotics. Serial vaginal swabs were obtained from diagnosis of PPROM through delivery and the sequencing of the V3–V5 region of the 16S rRNA gene was performed for all collected samples.

**Results.** The results show that Lactobacilli species were markedly decreased when compared to vaginal swabs collected from uncomplicated pregnancy subjects with a matched gestational time. *Prevotella* and *Peptoniphilus* were the most prevalent taxa in PPROM subjects at presentation. The vaginal microbiome of the PPROM subjects varied substantially intra- and inter-subjects. Several taxa were found to be significantly reduced during and after the antibiotic treatment: *Weeksella*, *Lachnospira*, *Achromobacter*, and *Pediococcus*. In contrast, *Peptostreptococcus* and *Tissierellaceae ph2* displayed a significant increase after the antibiotic treatment. However, the relative abundance of *Lactobacillus*, *Prevotella*, and *Peptoniphilus* was not substantially impacted during the hospitalization of the PPROM subjects. The deficiency of *Lactobacillus*, and constancy of known pathogenic species, such as *Prevotella* and *Peptoniphilus* during and after antibiotics, highlights the persistent dysbiosis and warrants further investigation into mitigating approaches.

**Discussion.** PPROM is responsible for one third of all preterm births. It is thought that subclinical infection is a crucial factor in the pathophysiology of PPROM because 25–40% of patients present signs of chorioamnionitis on amniocentesis. Here we sought to directly assess the bacterial content of the vagina and leaking amniotic fluid of subjects at presentation, throughout treatment and up until delivery, in order to search for common pathogens, microbiota changes, and microbial response to latency antibiotic treatment. We have found that the vaginal microbiome of PPROM subjects is highly variable and displays significant changes to treatment. However, the unchanging deficiency of *Lactobacillus*, and persistence of known pathogenic species, such as *Prevotella* and *Peptoniphilus* from presentation, through antibiotic treatment and up until delivery, highlights the persistent dysbiosis and warrants further investigation into mitigating approaches.

## INTRODUCTION

Preterm birth is the single leading cause of neonatal morbidity and mortality in the developed world, and despite efforts to identify causes, preventative strategies and treatment options, the incidence of preterm birth continues to rise in the United States (*Goldenberg et al., 2008*). Preterm premature rupture of membranes (PPROM) directly causes one third of all preterm births (*Mercer et al., 2000*). Subclinical infection likely plays a key role in the pathophysiology of PPROM and subsequent onset of preterm labor, as evidenced by the fact that 25–40% of patients with PPROM present signs of chorioamnionitis on amniocentesis (*Simhan & Canavan, 2005*). Furthermore, specific microbes associated with bacterial vaginosis (BV) such as *Gardnerella vaginalis* and *Mycoplasma hominis* have been linked to pregnancy complications, including PPROM, preterm birth (*Leitich et al., 2003*; *McDonald et al., 1991*; *Romero et al., 2002*; *Hay et al., 1994*), and intra-amniotic infections (*Mendz, Kaakoush & Quinlivan, 2013*). Attempts to prevent preterm birth by using antibiotics prior to the onset of labor have not been very efficacious (*Oliver & Lamont, 2013*; *Hauth et al., 1995*; *Brocklehurst et al., 2013*). This suggests that our current knowledge of the vaginal microbes associated with adverse pregnancy outcomes, mainly based on clinical and traditional culture techniques remains incomplete (*White et al., 2011*).

Recently, the advent of next generation sequencing techniques such as 16S rRNA gene hypervariable tag sequencing has allowed a more complete characterization of the vaginal microbial ecology also known as the microbiome. Studies using 16S rRNA gene technology have demonstrated a broad spectrum of microbial organisms not previously identified in the vagina using traditional culture techniques (*Fredricks, Fiedler & Marrazzo, 2005*; *Zhou et al., 2004*; *Ravel et al., 2010*). In studies of healthy non-pregnant women, the vaginal microbiome is characterized by five community profiles or subtypes most primarily dominated by a mixture of *Lactobacillus* species (*Ravel et al., 2010*). During normal

pregnancy, this dominance broadens and the microbial diversity undergoes a marked decrease with convergence toward a single subtype (*Aagaard et al., 2012*; *Romero et al., 2014b*; *Walther-António et al., 2014*). Although, the vaginal microbiome changes in relative abundance of taxa experienced by women that delivered at term has not been shown to be significantly different from women that delivered preterm (*Romero et al., 2014a*), higher absolute abundances of particular microorganisms such as *Leptotrichia/Sneathia*, *Mobiluncus spp.* (*Nelson et al., 2014*), and *Mycoplasma* (*Wen et al., 2014*) have been correlated to preterm deliveries. It is therefore intriguing that increased diversity in the vaginal microbiome in non-pregnant women correlates with clinical disease such as BV, which is also correlated with adverse pregnancy outcomes (*Oliver & Lamont, 2013*; *Oakley et al., 2008*; *Flynn, Helwig & Meurer, 1997*; *Hauth et al., 2003*). Despite this, little is known about the vaginal microbiome and amniotic fluid discharge associated with pregnancy complications such as PPROM.

After diagnosis of PPROM at a gestation of 34 weeks or less, the administration of antibiotics has been shown to prolong pregnancy latency and improve short-term neonatal outcome (*Kenyon, Boulvain & Neilson, 2013*; *Hutzal et al., 2008*). However, although outcomes are improved by antibiotic administration, the majority of women still enter spontaneous preterm labor soon after PPROM, and maternal and neonatal infectious morbidity is not eliminated (*Simhan & Canavan, 2005*; *Epstein, Parry & Strauss, 1998*; *Soraisham et al., 2009*). This suggests that there are residual microbial factors involved after broad-spectrum antibiotics. In order to directly characterize the vaginal and amniotic fluid discharge at a PPROM event, we have taken a metagenomic approach and used next generation sequencing techniques. We continued the sampling throughout the subjects' hospitalization up until delivery to capture the microbiome changes throughout the treatment and delivery events.

# MATERIALS AND METHODS

## Ethics statement

Subjects were consented under IRB #12-001675, which was reviewed and approved by the Mayo Clinic Institutional Review Board. All subjects provided written consent.

## Subjects enrollment

Here we report the results from 15 subjects enrolled upon admission for PPROM to the Mayo Clinic Hospital—Rochester. Inclusion criteria consisted of the following: age >18 years; no known pregnancy complications prior to admission; ability to provide written informed consent; willingness to participate in mandatory translational research component of the study; weight greater than 50 kg; and confirmed preterm rupture of membranes based on clinical criteria. Exclusion criteria consisted of the following: known immunodeficiency; chronic active viral infection (including HIV, HTLV and hepatitis); known autoimmune disease; solid organ or transplant recipient; and multiple gestations. Upon enrollment, participations were requested to fill out a questionnaire detailing sexual and reproductive health history and hygiene practices. Selected questionnaire data is

available in Table S1. Metadata from the questionnaires was stored using REDCap (*Harris et al., 2009*). Relevant medical information related to the time of labor and delivery is also included in Table S1. All subjects were managed according to the standard institutional protocol for PPROM. This included hospitalization, administration of a course of antenatal steroids if <34 weeks gestation on admission and a standardized course of latency antibiotics (12 subjects received ampicillin/amoxicillin and azithromycin; two subjects received clindamycin and azithromycin due to penicillin allergy; and one subject received penicillin alone due to presentation in active labor). Delivery was undertaken for spontaneous preterm labor, nonreassuring maternal or fetal status, clinical concern for infection, or when a gestational age of 34 weeks was reached.

## Sample collection

Dacron swabs were collected during sterile speculum exam performed by an obstetrician from the posterior vaginal fornix and placed in a Nucleic Acid Transport collection tube. After collection the samples were stored at −80 °C until processing. Vaginal samples were collected at the time of admission, every three days for the first two weeks of admission and weekly thereafter until delivery. An additional vaginal sample was collected at the time of delivery and a swab from the fetal surface of the placental membranes after delivery for 6 of the 15 subjects.

## Sample processing

Samples were thawed and centrifuged for 10 min at 10 000 $g$ to collect the bacterial cells, and the supernatant was discarded. Genomic DNA extraction was performed by using the MoBio Ultraclean Soil Kit (MoBio Laboratories, Inc., Carlsbad, CA); with the MP FastPrep (MP Biomedicals, Solon, OH) for 40 s at 6.0 m/s. Incubation period was done for a minimum of 30 min. After extraction the DNA content was measured using High Sensitivity Qubit (Life Technologies Corporation, Carlsbad, CA) with the results ranging from below detection to 23 ng/ul of DNA. The V3–V5 region of the 16S rRNA was then amplified using a two-step polymerase chain reaction (PCR) protocol. The primary amplification was done using the following recipe: 3 µl template DNA, 0.5 µl nuclease-free water, 1.2 µl 5× KAPA HiFi buffer (Kapa Biosystems, Woburn, MA), 0.18 µl 10 mM dNTPs (Kapa Biosystems, Woburn, MA), 0.3 µl DMSO (Fisher Scientific, Waltham, MA) , 0.12 µl ROX (25 µM) (Life Technologies, Carlsbad, CA), 0.003 µl 2,000 × SYBR Green, 0.12 µl KAPA HiFi Polymerase (Kapa Biosystems, Woburn, MA), 0.3 µl forward primer (10 µM), 0.3 µl reverse primer (µl). The following cycling conditions were used: 95 °C for 5 min, followed by 25 cycles of 98 °C for 20 s, 55 °C for 15 s, 72 °C for 1 min. The primers for the primary amplification contained both 16S-specific primers (357F and 926R), as well as adapter tails for adding indices in a secondary amplification. The primer sequences for the primary amplification were as follows (16S-specific sequences in bold):
V3F_Nextera:
TCGTCGGCAGCGTCAGATGTGTATAAGAGACAG**CCTACGGGAGGCAGCAG**
V5R_Nextera:
GTCTCGTGGGCTCGGAGATGTGTATAAGAGACAG**CCGTCAATTCMTTTRAGT**

Next, these amplicons were diluted 1:100 in sterile, nuclease-free water, and a second PCR reaction was set up to add the Illumina flow cell adapters and indices. The secondary amplification was done using the following recipe: 5 µl template DNA, 1 µl nuclease-free water, 2 µl 5 × KAPA HiFi buffer (Kapa Biosystems, Woburn, MA), 0.3 µl 10 mM dNTPs (Kapa Biosystems, Woburn, MA), 0.5 µl DMSO (Fisher Scientific, Waltham, MA), 0.2 µl KAPA HiFi Polymerase (Kapa Biosystems, Woburn, MA), 0.5 µl forward primer (10 µM), 0.5 µl reverse primer (µl). The following cycling conditions were used: 95 °C for 5 min, followed by 10 cycles of 98 °C for 20 s, 55 °C for 15 s, 72 °C for 1 min, followed by a final extension at 72 °C for 10 min. The indexing primers are as follows (X marks the positions of the 8 bp indices):

Forward indexing primer:

**AATGATACGGCGACCACCGA**GATCTACACXXXXXXXXTCGTCGGCAGCGTC

Reverse indexing primer:

**CAAGCAGAAGACGGCATACGA**GATXXXXXXXXGTCTCGTGGGCTCGG

The products of the amplification were quantified using a PicoGreen dsDNA assay (Life Technologies, Carlsbad, CA), and the samples were normalized, pooled, and approximately 1 µg of material was concentrated to 10 µl using 1.8 × AMPureXP beads (Beckman Coulter, Brea, CA). The pooled sample was then size selected at 723 bp ± 20% on a Caliper XT DNA 750 chip (Caliper Life Science, Hopkinton, MA). The size-selected material was cleaned up using AMPureXP beads, and eluted in 20 µl of EB buffer (10 mM Tris–HCl, pH 8.5). The final pooled sample was quantified using the PicoGreen dsDNA assay, and analyzed using an Agilent Bioanalyzer High Sensitivity Chip (Agilent Technologies, Santa Clara, CA). Finally, the sample pool was diluted to 2 nM based on the PicoGreen measurements, and 10 µl of the 2 nM pool was denatured with 10 µl of 0.2 N NaOH, diluted to 8 pM in Illumina's HT1 buffer, spiked with 20% phiX, heat denatured at 96 °C for 2 min, and sequenced using a MiSeq 600 cycle v3 kit (Illumina, San Diego, CA).

## Processing of controls

Control samples consisted of five "empty collection" replicates. In brief, five swabs were immersed in the collection buffer and DNA was extracted, amplified, and sequenced using the same procedures and reagents used for the PPROM samples. In addition, three other controls were performed: (1) one extraction with only the collection buffer was performed (no swab), (2) a PCR negative control was sequenced, (3) a positive control of a pure isolate of *Campylobacter jejuni*. There was no detectable amplification in the negative controls by qPCR. The positive control (*Campylobacter jejuni*) was carried forward to sequencing as well to guarantee that the negative controls were sequenced at comparable depth. The eight control samples were spiked into a MiSeq run at a comparable per-sample concentration to that of the PPROM sample MiSeq run. As expected, the extraction and water negative controls yielded much lower numbers of reads (between 0.0084% and 0.0406% of the total reads in the run).

## Sequencing processing and outcome

To remove the 5'-Adapter-primer sequence on forward reads which has a total length of 58 bps, for each forward read, we took 60 bps from the 5' end and BLAST against the 5'-Adapter-primer sequence allowing one base mismatch, and the matched region of the reads were removed if present at the beginning of the sequence read. To remove the 5'-Adapter-barcode-primer sequence on reverse reads which has a total length of 65 bps, we first generated a group of sequences, through a combination of different barcodes and wobbly bases. There are a total of 73 samples taken from 15 subjects, each sample with a unique barcode. And there are two wobbly bases M and G in the sequence, each of which denotes an alternative base of A or C and A or G respectively. Thus the total number of sequence combinations is $77 \times 2 \times 2 = 292$. We created a BLAST database from the 292 sequences. Then, for each reverse read, we took 67 bps from the 5' end and BLAST against the created database allowing one base mismatch except for the barcode region. The matched region of each read was removed if present at the beginning of the sequence read and the samples were de-multiplexed by the matched barcode on each read. Due to the low number of R2 samples passing quality control (at least 187 bp per sequence read and a minimum of 2,000 sequence reads per sample), only R1 reads were used for analysis. A total of 5,575,178 R1 sequence reads (5,527–174,479 sequence reads per sample) passed quality control.

## Sequence analysis

Sequence reads were aligned with our own custom multiple alignment tool known as the Illinois-Mayo Taxon Operations for RNA Dataset Organization (IM-TORNADO) that merges paired end reads into a single multiple alignment and obtains taxa calls (*Sipos et al., 2010*). Operational taxonomic units were clustered using UPARSE (*Edgar, 2013*). Further processing for visualization and statistical analysis was performed using QIIME (*Caporaso et al., 2010*). To identify differential abundant bacteria genera, we fit a linear regression model to the square-root transformed genus proportion data. We used the *t*-statistic as the test statistic. To address the non-normality of the outcome variable as well as within-subject correlation, we used permutation to assess the statistical significance. Permutation was performed 1,000 times and was constrained within the subject to retain the original correlation structure. We also fit a generalized mixed effects model to the genus presence/absence data using PQL method, assuming a random intercept for each subject. Statistical significance was assessed based on Wald test. False discovery control (B-H procedure) was used for correcting multiple testing. Sequences are publicly available at SRA, study accession: SRP061714. The taxonomic assignments are provided in Table S2.

## RESULTS

Our cohort consisted of 15 subjects (Table 1; age range: 19–37 years old; mean ± standard deviation: 29 ± 4 years old). The majority were Caucasian (14), and one was East Asian. Within the cohort, there were no complications specific to pregnancy prior to or subsequent to diagnosis of PPROM including no diagnoses of gestational diabetes or

**Table 1** Demographics and clinical data for all subjects enrolled.

|  | Range | Mean (SD) |
|---|---|---|
| Age (years) | 19–37 years | 29 years (±4) |
| Gestational age (weeks) |  |  |
| On admission | 23 1/7 to 34 5/7 weeks | 30 5/7 weeks (±3.6) |
| At delivery | 28 1/7 to 34 6/7 weeks | 32 5/7 weeks (±1.9) |
| Latency period (days) | 0–58 days | 15 days (±17.4) |
| Vaginal samples collected (no.) | 1–11 | 4 (±3.2) |
| Gravid status | 1–4 | 2 (±1) |
| Parity | 0–3 | 1 (±1)–33% Nulliparous |
| Racial/ethnic background (subjects) |  |  |
| White or Caucasian—14 (93%) |  |  |
| Near East Asian—1 (17%) |  |  |
| Type of delivery |  |  |
| NSVD—10 (67%) |  |  |
| C-section—5 (33%) |  |  |
| Evidence of chorioamnionitis |  |  |
| 4 (27%) |  |  |

**Notes.**
SD, Standard Deviation; Gravid Status, Number of prior pregnancies; Parity, Number of prior deliveries; NSVD, Normal Spontaneous Vaginal Delivery.

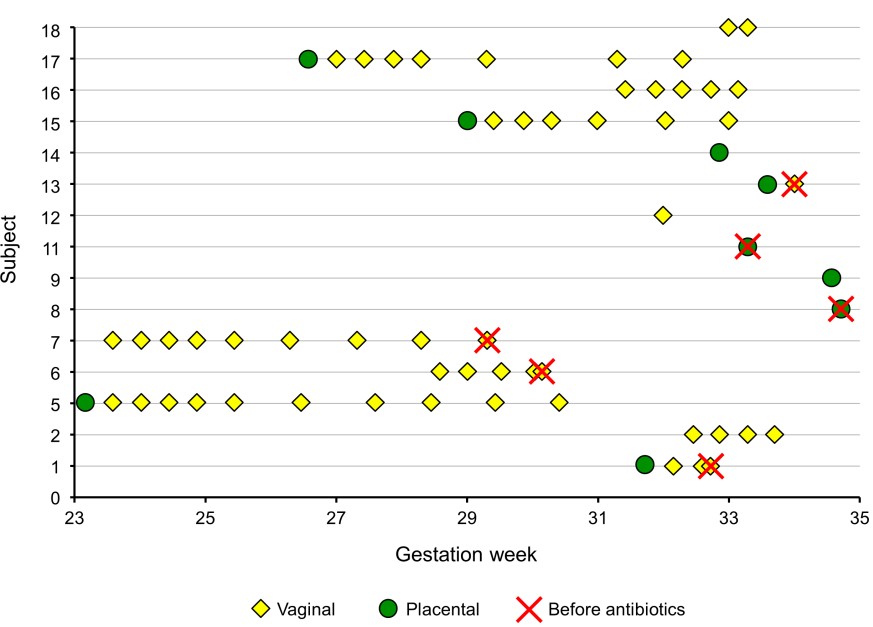

**Figure 1** Samples collected and analyzed in the course of the study.

hypertensive disorders of pregnancy. The cohort included 10 parous and 5 nulliparous women, with four subjects having a history of preterm delivery with a previous pregnancy. From these 15 subjects, a total of 61 vaginal samples and 6 placental samples were collected (Fig. 1; selected metadata associated with each subject can be found in Table S1).

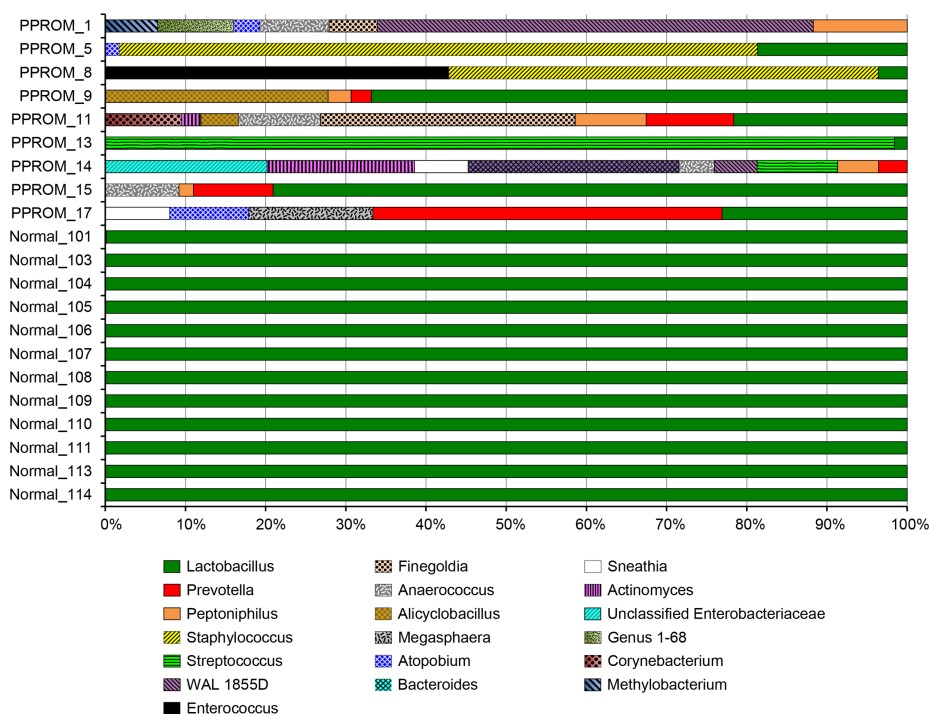

**Figure 2 Comparison of relative abundance of taxa at the genus level between vaginal swabs of subjects at the time of presentation at the emergency room with PPROM before the administration of antibiotic treatment (9 subjects) and vaginal swabs from subjects that underwent an uncomplicated pregnancy at approximately 29 weeks of gestation (12 subjects).** The lack of lactobacilli dominance in PPROM subjects is apparent, as is the inter-individual variation. Only taxa at >1% relative abundance in the PPROM subjects are shown for graphical clarity.

## PPROM before antibiotic treatment

In order to assess the näive maternal microbiome at time of presentation with PPROM, we examined the admission samples from nine subjects that were obtained before the administration of antibiotics and compared them with those of a companion study, where 12 women experiencing an uncomplicated pregnancy were sampled longitudinally (*Walther-António et al., 2014*). Using gestational age-matched samples from the companion study we compared the vaginal microbiome taxonomic assignments to the PPROM subjects. The microbiome in the PPROM subjects exhibited a diversity of taxa that was in stark contrast to the normal pregnancy subjects (Fig. 2), and was marked by an underrepresentation of lactobacilli (Fig. 3). Overall, the most common pathogens detected in the PPROM subjects before the administration of antibiotics were *Prevotella* and *Peptoniphilus* (Fig. 4).

## PPROM during and after antibiotic treatment

The administration of antibiotics did not significantly impact the relative abundance of *Lactobacillus*, *Prevotella*, or *Peptoniphilus* during or after the antibiotic treatment (Tables 2 and 3). However, during the administration of the treatment, *Weeksella, Lachnospira, Achromonacter* and *Pediococcus* showed a significant decrease, while *Peptostreptococcus* and *Tissierellaceae ph2* showed a significant increase (Table 2 shows presence/absence

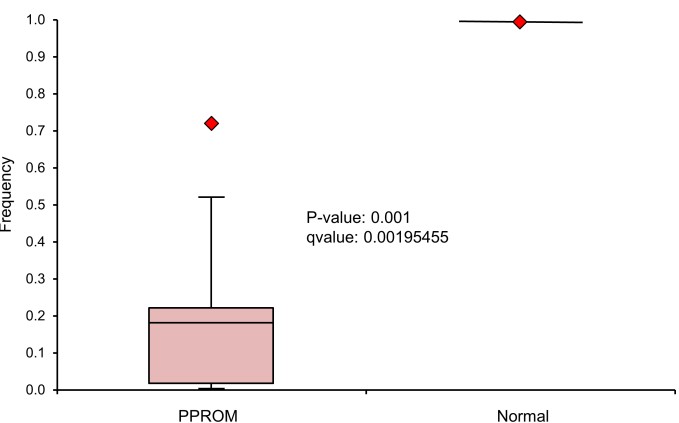

**Figure 3 Boxplot representing the *Lactobacillus* frequency in the vaginal swabs of 9 PPROM subjects at presentation before the administration of antibiotic treatment.** Normal is represented by the vaginal swabs of 12 subjects with uncomplicated pregnancies at approximately 29 weeks of gestation. Statistical significance found through linear regression with permutation.

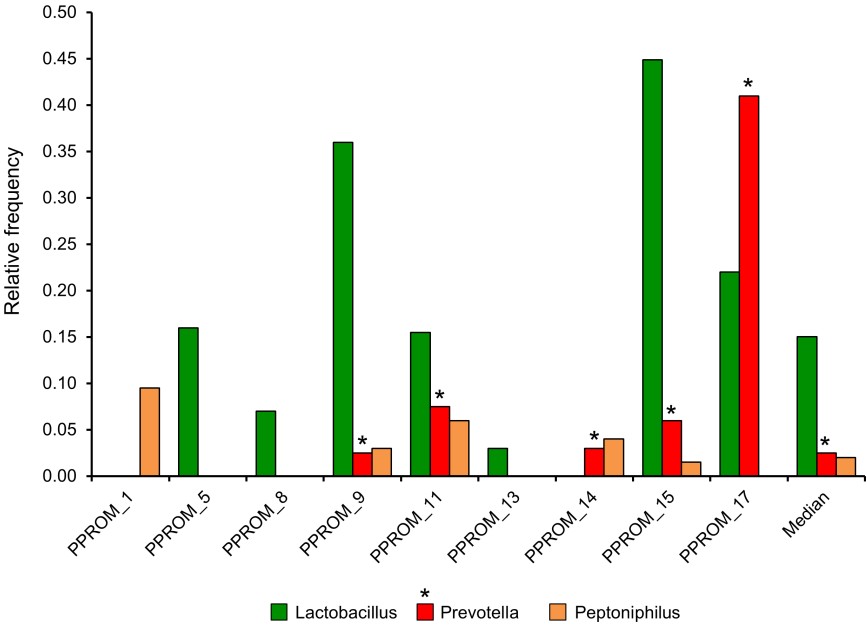

**Figure 4 Summary (Median >1% relative abundance) of taxa at the genus level from all PPROM subjects at presentation before the administration of antibiotic treatment (9 subjects).**

significance most powerful for low abundance taxa, Table 3 shows proportion significance more powerful for high abundance taxa). We also compared the Bray-Curtis (BC) and UniFrac distances (with similar results) to address relevant questions. One of them was whether there was a statistical difference in the vaginal microbiome of subjects that underwent a cesarian-section (C-section) when compared to subjects that had a normal spontaneous vaginal delivery (NSVD). We have 5 subjects that underwent a C-section and 10 with a NSVD. PERMANOVA based on BC distance produced a *p*-value of 0.23. Therefore, we do not have enough supporting evidence that the delivery mode affects

**Table 2  Taxa displaying significant changes before, during, and after antibiotic treatment.[a]**

| Taxa (Phylum, Classe, Order, Family, Genus) | Before vs. during antibiotic treatment | | | During vs. after antibiotic treatment | | | Before vs. after antibiotic treatment | | |
|---|---|---|---|---|---|---|---|---|---|
| | value | p-value[b] | q value[c] | value | p-value[b] | q value[c] | value | p-value[b] | q value[c] |
| Bacteroidetes, Flavobacteriia, Flavobacteriales, Weeksellaceae, **Weeksella** | −28.3 | 0.9998 | 1.0000 | −2.5 | 0.0054 | 0.1482 | 25.8 | 0.9999 | 1.0000 |
| Firmicutes, Clostridia, Clostridiales, Lachnospiraceae, **Lachnospira** | −2.0 | 0.0743 | 0.6956 | −2.3 | 0.0048 | 0.1482 | −0.3 | 0.7712 | 1.0000 |
| Proteobacteria, Betaproteobacteria, Burkholderiales, Alcaligenaceae, **Achromobacter** | −1.2 | 0.2095 | 0.7810 | −2.7 | 0.0033 | 0.1482 | −1.5 | 0.1894 | 0.7564 |
| Bacteroidetes, Bacteroidia, Bacteroidales, Prevotellaceae, **Prevotella** | 28.1 | 0.9999 | 1.0000 | 28.3 | 0.9999 | 1.0000 | 0.2 | 1.0000 | 1.0000 |
| Firmicutes, Clostridia, Clostridiales, Tissierellaceae, **Peptoniphilus** | 0.7 | 0.5395 | 0.8650 | 0.2 | 0.7926 | 0.9266 | −0.6 | 0.6469 | 0.9301 |

Notes.

[a] Results achieved using a generalized mixed effects model to the present/absent taxa using the Penalized Quasi-likelihood (PQL) method, assuming a random intercept for each subject.

[b] Statistical significance was assessed based on Wald test.

[c] False discovery control Benjamini–Hochberg (B-H procedure) was used for correcting multiple testing ($q$ value $< 0.2$ (false discovery rate) considered significant).

the microbiome. We further enquired whether parity was a determining factor for the vaginal microbiome. There are 10 parous subjects (parity $> 0$) and 5 nulliparous subjects (parity $= 0$). PERMANOVA based on BC distance produced a $p$-value of 0.20. Therefore, we do not have enough supporting evidence that parity influences the vaginal microbiome.

## Placental microbiome

To assess the placental microbiome, we compared the six placental samples with the matched maternal vaginal samples obtained at the time of delivery. The placental microbiome was statistically distinct from the vaginal microbiome (Bray Curtis distance $p$-value 0.08—PERMANOVA, within subject 1,000 permutations). However, given the limited amount of samples, there was insufficient statistical power to identify differential taxa. Placental samples showed high individual variability and weak correlation with the maternal vaginal microbiome (Fig. 5). The most prevalent taxa recovered were *Alicyclobacillus* and *Corynebacterium*. In order to determine if the detected placental microbiome could be the result of contamination from the vaginal microbiome we performed a permutation test and compared the BC and Unifrac distances. There are six subjects that have matched placental and vaginal samples. Two subjects underwent a C-section and four subjects had a NSVD. We do not see that the placental swabs from the vaginal deliveries are significantly closer to the vaginal swabs than the placental swabs from C-sections ($p = 0.53$). However, one placental swab from a NSVD subject (PPROM_7) has potential contamination indicated by a very small distance (0.48) when compared to the range among the remaining NSVD distances (0.97–0.99) and C-sections (0.88–0.91). We

**Table 3 Significant taxa shifts before, during, and after antibiotic treatment.[a]**

| Taxa (Phylum, Classe, Order, Family, Genus) | Before Mean | During Mean | After Mean | Before vs. during p-value[b] | During vs. after p-value[b] | Before vs. after p-value[b] | Before vs. during q value[c] | During vs. after q value[c] | Before vs. after q value[c] | Signal |
|---|---|---|---|---|---|---|---|---|---|---|
| Firmicutes, Bacilli, Lactobacillales, Lactobacillaceae, **Pediococcus** | 1.9E−03 | 4.1E−03 | 1.6E−05 | 0.299 | 0.004 | 0.002 | 0.84 | 0.084 | 0.084 | Decrease |
| Proteobacteria, Betaproteobacteria, Burkholderiales, Alcaligenaceae, **Achromobacter** | 4.3E−04 | 1.8E−02 | 3.4E−06 | 0.267 | 0.003 | 0.018 | 0.84 | 0.084 | 0.308 | Decrease |
| Firmicutes, Clostridia, Clostridiales, Peptostreptococcaceae, **Peptostreptococcus** | 2.1E−03 | 2.7E−04 | 1.9E−02 | 0.247 | 0.001 | 0.001 | 0.84 | 0.042 | 0.084 | Increase |
| Firmicutes, Clostridia, Clostridiales, Tissierellaceae, **ph2** | 1.5E−05 | 0.0E+00 | 1.3E−04 | 0.69 | 0.001 | 0.019 | 0.92 | 0.042 | 0.308 | Increase |
| Firmicutes, Clostridia, Clostridiales, Tissierellaceae, **Peptoniphilus** | 2.9E−02 | 1.0E−02 | 4.2E−03 | 0.052 | 0.691 | 0.4 | 0.546 | 0.921 | 0.781 | – |
| Firmicutes, Bacilli, Lactobacillales, Lactobacillaceae, **Lactobacillus** | 2.1E−01 | 1.5E−01 | 1.4E−01 | 0.021 | 0.853 | 0.618 | 0.546 | 0.940 | 0.881 | – |
| Bacteroidetes, Bacteroidia, Bacteroidales, Prevotellaceae, **Prevotella** | 7.3E−02 | 1.3E−01 | 1.6E−01 | 0.457 | 0.782 | 0.661 | 0.868 | 0.940 | 0.881 | – |

**Notes.**

[a] A Linear regression model was fit to the square-root transformed genus proportion data. To address the non-normality of the outcome variable as well as within-subject correlation, permutation (1,000 times) was used to assess the statistical significance. Permutation was constrained within each subject to retain the original correlation structure.

[b] Statistical significance was assessed by permutation.

[c] False discovery control Benjamini–Hochberg (B-H procedure) was used for correcting multiple testing ($q$ value $< 0.2$ (false discovery rate) considered significant).

also looked into whether the delivery vaginal microbiome was the most similar to the placental swab or if the placental swab was rather closer to a previous vaginal swab from the same subject. There are four subjects with multiple vaginal swabs and a placental swab. By comparing the distance from the placental swab to the delivery swab, to the distance from the placental swab to other vaginal swabs within the same subject, we see a trend that the placental swab is more similar to the last delivery sample ($p = 0.078$, permutation test, shuffling the vaginal swabs within the same subject).

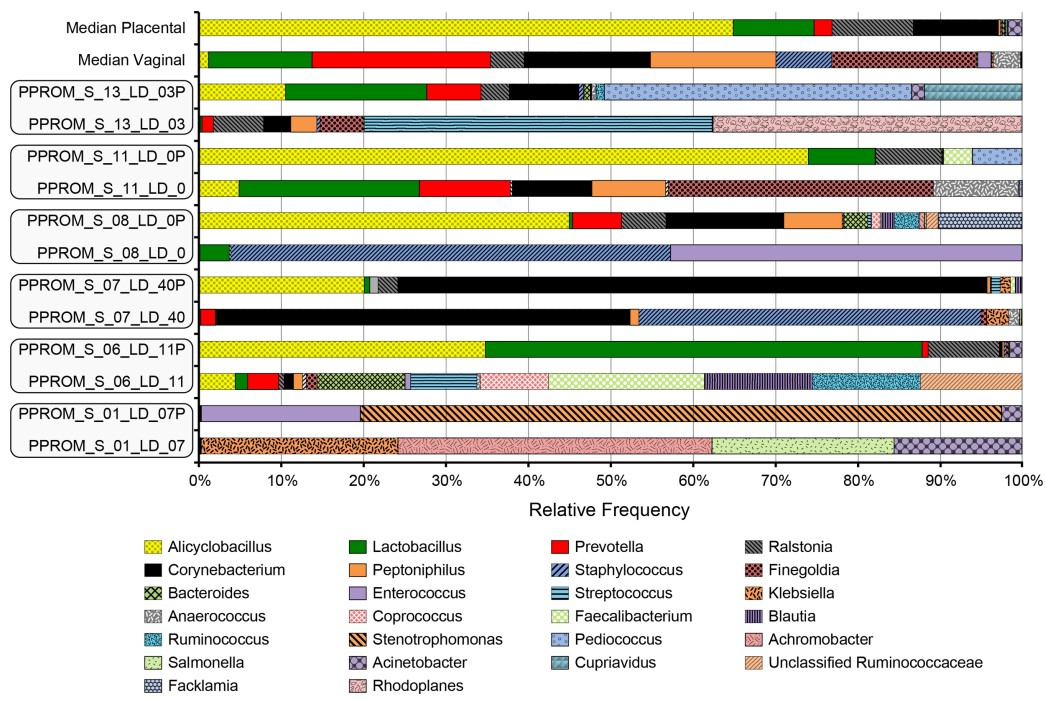

**Figure 5** **Placental and maternal microbiome taxa variation.** S, Subject; LD, Latency day; P, Placental sample.

## Controls

The sequencing of our controls—Negative PCR control, five quality control (QC) replicates (empty swab collections) and the DNA extraction and amplification of the TE buffer used for the sample collection—showed very little overlap with the major taxa seen in the study samples (Table S3).

## DISCUSSION

### Principal findings of the study

The analysis of the vaginal microbiome of 15 PPROM subjects from presentation, throughout their hospital stay and treatment, and up until delivery, shows that the microbiome is dysbiotic and highly variable between subjects since presentation. The antibiotic treatment administered for the condition did not eliminate the presence of pathogenic species such as *Prevotella* and *Peptoniphilus*, which remained until the time of delivery, as did the deficiency in lactobacilli species.

Our results also show that the vaginal microbiota of pregnant subjects with PPROM is distinct from that of subjects undergoing an uncomplicated pregnancy. The findings indicate that the vaginal microbiome after PPROM is more diverse and dynamic than previously observed during normal pregnancies (*Walther-António et al., 2014*). Of specific interest is the difference in lactobacilli abundance in PPROM. Lactobacilli acidify their environment and limit diversity through a process of niche expansion—allowing for the dominance of lactobacilli in most vaginal microbiota observed to date. The loss of

lactobacilli, along with many other factors, may place a key role in the destabilization of the PPROM-associated vaginal microbiome. This depletion may be due to the displacement either by the constant leakage of alkaline amniotic fluid or may precede the PPROM event. It is also worth emphasizing that although the swabs were physically placed in the posterior fornix for the sampling, this was likely to contain varying amounts of amniotic fluid in PPROM subjects. Once there is a rupture of the membranes, amniotic fluid continues to leak from the cervix and when a patient is in the supine position the posterior fornix is where that fluid will collect. The patients remain hospitalized until delivery, and the samples were obtained in the patient's hospital bed.  So unless the patient had been ambulating just prior to sampling, there will be some amniotic fluid pooled in the posterior fornix. Once lactobacilli are displaced, opportunistic bacteria may occupy the available niche leading to an unbalanced microbial ecology (dysbiosis) with potentially negative consequences for health. The transient nature of the PPROM microbiome is consistent with the idea of an ecology that is out of equilibrium. *Prevotella* and *Peptoniphilus* emerge as taxa of particular interest in PPROM given their prevalence at presentation and persistence throughout treatment, evidence of prominent role in persistent bacterial vaginosis and preterm labor (*Marrazzo et al., 2008*; *Smayevsky et al., 2001*; *Wang et al., 2013*; *Mikamo et al., 1999*) and display of broad-spectrum antibiotic resistance (*Sherrard et al., 2013*; *Tanaka et al., 2006*). The highly variable placental microbiome recovered and the weak correlation with the maternal microbiome raises the possibility that the uterine microbiome may be an independent driver of the placental microenvironment. The formation of the cervical mucus plug as early as seven weeks of gestation provides a mechanical and chemical barrier between the vaginal and uterine environments (*Hein et al., 2002*). The influence of the vaginal microbiome in the uterine microbial niche is therefore anticipated to be reduced for as long as the mucus plug is intact. The fact that in this study *Prevotella* was a prominent pathogen found in the vaginal fluid discharge and placental membranes indicates a very likely role in the etiology of PPROM. The known association of *Prevotella* with bacterial vaginosis (*Hillier et al., 1993*) and preterm labor (*Holst, Goffeng & Andersch, 1994*) strengthen this possibility. It is however, important to note that preterm labor and PPROM are distinct conditions. While PPROM often leads to preterm labor, preterm labor can be caused by a multitude of conditions unrelated to PPROM. It is therefore not surprising that preterm labor studies (*Romero et al., 2014a*; *Mendz, Kaakoush & Quinlivan, 2013*) may or may not align with our findings.

Our study is limited by the small number of patients. Another limitation is that the vaginal swabs collected from PPROM subjects likely contained both amniotic fluid and vaginal fluid. We are unable to determine the proportion of amniotic fluid in each sample, which is expected to vary with each collection. By contrast, the vaginal swabs collected from uncomplicated pregnancy subjects are not anticipated to have had a significant presence of amniotic fluid.

Despite the low amount of microbial DNA present in the samples, we were able to rule out that the relevant taxa in this study could be the result of contamination. As shown

in Table 3 we were able to find several well-known contaminants (*Salter et al., 2014*), but not the taxa relevant to our study with one exception, *Corynebacterium*. This taxa was predominant in the placenta of our PPROM subjects, which supports previous findings in the placenta of preterm deliveries (*Oh et al., 2010*). It is possible that its presence may be overemphasized in this study due to its detection in the collection buffer, but it is undoubtedly present in the study samples as well.

## CONCLUSIONS

The deficiency in lactobacilli species and persistence of known pathogenic species at admission, and during and after antibiotics, highlights a marked dysbiosis with high individual variability. An interesting area of future research would be to assess the changes in the microbiome noted in our study as a potential causative factor predating diagnosis of PPROM by longitudinal assessment of the vaginal microbiome in pregnant women at high risk for preterm premature rupture of membranes.

## ACKNOWLEDGEMENTS

The authors wish to acknowledge clinical, diagnostic and clerical staff at Mayo Clinic.

### Funding

This project was funded by the Center for Individualized Medicine, Mayo Clinic. The funders had no role in study design, data collection and analysis, decision to publish, or preparation of the manuscript.

### Grant Disclosures

The following grant information was disclosed by the authors:
Center for Individualized Medicine, Mayo Clinic.

### Competing Interests

DC and NC have licensed intellectual property with Whole Biome, Inc. Jun Chen is an Academic Editor for PeerJ. The authors declare there are no competing interests.

### Author Contributions

- Elizabeth A. Baldwin conceived and designed the experiments, wrote the paper, reviewed drafts of the paper.
- Marina Walther-Antonio conceived and designed the experiments, performed the experiments, analyzed the data, wrote the paper, prepared figures and/or tables, reviewed drafts of the paper.
- Allison M. MacLean and Daryl M. Gohl performed the experiments, reviewed drafts of the paper.
- Kenneth B. Beckman performed the experiments, contributed reagents/materials/analysis tools, reviewed drafts of the paper.
- Jun Chen analyzed the data, prepared figures and/or tables, reviewed drafts of the paper.

- Bryan White and Douglas J. Creedon conceived and designed the experiments, reviewed drafts of the paper.
- Nicholas Chia conceived and designed the experiments, analyzed the data, contributed reagents/materials/analysis tools, wrote the paper, reviewed drafts of the paper.

### Human Ethics

The following information was supplied relating to ethical approvals (i.e., approving body and any reference numbers):

Mayo Clinic Institutional Review Board: #12-001675.

### DNA Deposition

The following information was supplied regarding the deposition of DNA sequences:

Sequences are publicly available at SRA: SRP061714.

### Supplemental Information

Supplemental information for this article can be found online at http://dx.doi.org/10.7717/peerj.1398#supplemental-information.

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
