# Peer review of "Persistent microbial dysbiosis in preterm premature rupture of membranes from onset until delivery"

_PeerJ, doi:10.7717/peerj.1398_

## Round 0.1 · original submission · Major Revisions

Although the reviewers both complimented the work done by the authors they also raised serious concerns regarding the results and their interpretation by the authors as well as the quality of the discussion of these results

These comments needs to be addressed in order for this manuscript to be consider for publication

Reviewer 1 ·

Basic reporting

No Comments

Experimental design

No Comments

Validity of the findings

No Comments

Additional comments

This is an interesting study that aimed to investigate the vaginal microbiome and amniotic
fluid discharge in PPROM to identify pathogens and the microbial response to antibiotics treatment. In good accordance with earlier studies it was found that the vaginal microbiome in PPROM was variable and deficient in Lactobacillus, and displayed some changes to treatment.
However, there was no change in the deficiency of Lactobacillus and there was the persistence of
known pathogens (e.g. Prevotella, Peptoniphilus). The authors concluded that this finding
highlights a persistent dysbiosis in PPROM in spite of the antibiotics treatment, and warrants further investigation.

The study is well designed and executed, and the paper nicely presents the complex data. The introduction and discussion give an overall good overview of the topic and put the findings into context. Because of these and the importance of the findings, the reviewer suggests this paper to be published in PeerJ. However, there are some issues which necessitate revision:

- A Table with demographis and clinical characteristics needs to be included.
- How many patients went under C-section? Was there any difference in microbial diversity between C-section and vaginal delivery patients?
- Is it possible that the ‘placental microbiome’ on the surface of the placenta was only contamination from vaginal delivery?
- Line 228: ‘marked’ is used twice – it would be better to rephrase the sentence.
- Was there any difference in the microbiomes between nulliparous and parous women?
- In Table 1, some cells are also highlighted presenting non-significant data.
- Figure 1 legend is missing.
- The formatting of the references is often incorrect.

·

Basic reporting

9/16/2015

To
Prof. Offer Erez
Academic Editor
PeerJ

Dear Prof. Erez

Enclosed please find a review of the manuscript entitled "PERSISTENT MICROBIAL DYSBIOSIS IN PRETERM PREMATURE RUPTURE OF MEMBRANES FROM ONSET TO DELIVERY” which I am recommending for publication in PeerJ after a minor revision. I have prepared a summary of the study and a list of issues that the authors may want to address.

Regards,

Salvatore Andrea Mastrolia M.D.
Department of Obstetrics and Gynecology
Azienda Ospedaliera Universitaria Policlinico di Bari
University of Bari “Aldo Moro”, Bari, Italy
Piazza Giulio Cesare 11, 70123
Bari, Italy
Tel +390805593583, Fax +390805592228
International Fellowship Program
Maternal Fetal Medicine Unit
Department of Obstetrics and Gynecology “B”
Soroka University Medical Center
Ben Gurion University of the Negev
BeerSheva, Israel
E-mail mastroliasa@gmail.com

Peer J Manuscript ID: #2015:08:6453:0:0:REVIEW
Title: " PERSISTENT MICROBIAL DYSBIOSIS IN PRETERM PREMATURE RUPTURE OF MEMBRANES FROM ONSET TO DELIVERY”

Summary: Baldwin et al performed a prospective study to analyze the vaginal microbiome in patients with preterm premature rupture of membranes (PPROM), comparing it with uncomplicated pregnancy. They aimed to evaluate and characterize the changes in the microbiome and the influence of antiobiotic treatment in patients with PPROM. Samples were obtained in 15 patients through vaginal swabs at the time of admission and during the time of hospitalization, and placental swab was performed in six patients at delivery. A dysbiosis consisting in a decreased presence of Lactobacilli and increased presence of species such as Prevotella and Peptoniphilus at presentation of patients with PPROM is reported. Of interest, this condition was not consistently modified during antimicrobial therapy.
In conclusion, the authors suggest such dysbiosis as a potential mechanism for PPROM and preterm parturition.

Experimental design

Please read General comments for the Author

Validity of the findings

Please read General comments for the Author

Additional comments

General comments: I read with interest the manuscript from Baldwin et al and I appreciated their approach based not only on the known fact of the association among infection, PPROM, and preterm parturition, but that a misbalance in the microbial vaginal and, maybe, intrauterine condition, may trigger these obstetrical syndromes. Here is a list of minor concerns that the authors may want to address:

1) The authors are discussing the potential effect of dysbiosis in the development of PPROM and preterm labor and this seems to be the strong subject on which they designed and developed their study. I must say this does not come out clear to the reader in the introductive paragraph. Data regarding the impact of preterm labor and preterm parturition, as well as the association between PROM and the onset of labor are important and need to be provided. Moreover, I would focus introduction describing the microbial vaginal balance and its relation with normal uncomplicated pregnancy in order to justify the importance of a study investigating the role of a misbalance of such a microbiome.

2) I think there are interesting papers that the authors may want to discuss and compare to their findings, since this would increase the value of their findings and give a more complete picture of this interesting field of research in Obstetrics. I am referring to the manuscript from Roberto Romero et al comparing vaginal microbiota in women with spontaneous preterm labor vs. those delivering at term (Microbiome. 2014 May 27;2:18. doi: 10.1186/2049-2618-2-18. eCollection 2014) and a review article published in 2013 by Mendz et al (Front Cell Infect Microbiol. 2013; 3: 58) dealing with bacterial aetiological agents of intra-amniotic infections and preterm birth in pregnant women.



Specific comments

Title: Appropriate.

Abstract: I would suggest the authors to focus their abstract on the topic of their study that is the dysbiosis in vaginal microbiome. The actual version of the abstract does not attract the interest of the reader.

Introduction: Please refer to my general comments.

Materials and methods: The description of the methods is precise and extensive.

Results: This section is well written and clearly understandable to the reader. I only have one concern regarding the fact that placental swab was compared to vaginal swab performed at the moment of delivery. Since the process of preterm parturition is triggered by changes that start a certain time before the clinical observation of the process of labor, and even more in case of PPROM, maybe it would be interesting to compare placental swab to those obtained during pregnancy affected with PPROM.

Discussion:
1) I would suggest the authors to start the Discussion section with a paragraph named "Principal findings of the study". Otherwise the reader finds himself reading the entire discussion in order understand which is novel. Since this study does have something novel, it should be clearly stated immediately.
2) The limitations of the study are well described and the authors themselves recognize the impossibility to determine the amount and the contribution of amniotic fluid to the analysis of the vaginal microbiome obtained from vaginal swabs performed at the posterior fornix. Moreover, on page#15, lines 282-284 the authors state that "The fact that in this study Prevotella was a prominent pathogen found in the vaginal, amniotic fluid discharge and placental membranes indicates a very likely role in the etiology of PPROM". I agree with them regarding vaginal and placenta but I think there is no chance to state that Prevotella was a prominent pathogen in amniotic fluid discharge.
3) Page#15, lines 278-279: "Although the vaginal microbiome likely exerts a strong influence in the uterine microbiome in the initial stages of pregnancy, ...". Page#15, lines 284-285: " The known association of Prevotella with bacterial vaginosis and preterm labor strengthen this possibility". These sentences should be referenced.
4) Please read my general comments regarding the discussion of recent papers to be included in the Discussion section

Conclusions: Adequate.

Tables and supplementary material: Nothing to report.

---

## Round 0.2 · accepted · Accept

The authors have addressed all the concerns raised by the reviewers and both of them found the manuscript suitible for publication

Reviewer 1 ·

Basic reporting

No Comments

Experimental design

No Comments

Validity of the findings

No Comments

Additional comments

The authors have performed the requested changes to the manuscript. Thus, I suggest accepting the paper for publication.

·

Basic reporting

20/10/2015

To
Prof. Offer Erez
Academic Editor
PeerJ

Dear Prof. Erez,
Enclosed please find a review of the manuscript entitled "PERSISTENT MICROBIAL DYSBIOSIS IN PRETERM PREMATURE RUPTURE OF MEMBRANES FROM ONSET TO DELIVERY” which I am recommending for publication in PeerJ.

Regards,

Salvatore Andrea Mastrolia M.D.
Department of Obstetrics and Gynecology
Azienda Ospedaliera Universitaria Policlinico di Bari
University of Bari “Aldo Moro”, Bari, Italy
Piazza GiulioCesare 11, 70123
Bari, Italy
Tel +390805593583, Fax +390805592228
International Fellowship Program
Department of Obstetrics and Gynecology “B”
Soroka University Medical Center
Ben Gurion University of the Negev
Beer Sheva, Israel
E-mail mastroliasa@gmail.com

Experimental design

none

Validity of the findings

none

Additional comments

Peer J Manuscript ID: # 2015:07:5918:1:0:REVIEW
Title: "PERSISTENT MICROBIAL DYSBIOSIS IN PRETERM PREMATURE RUPTURE OF MEMBRANES FROM ONSET TO DELIVERY”


Comments for the author
I have reviewed a previous version of this manuscript and assigned to the authors minor revision comments. I am fully satisfied with the answers provided by the authors and I think this study might be an excellent start for future research projects in the field or preterm parturition and obstetrical syndromes. Therefore, I recommend this manuscript for publication in PeerJ.